

# Restudy of shoulder motion in the theropod dinosaur *Mononykus olecranus* (Alvarezsauridae)

Philip J. Senter

Department of Biological and Forensic Sciences, Fayetteville State University, Fayetteville, North Carolina, United States

## ABSTRACT

**Background:** Range of motion in the forelimb of the Upper Cretaceous theropod dinosaur *Mononykus olecranus*, a member of the family Alvarezsauridae, has previously been investigated. However, the method used to investigate range of motion at the shoulder in *M. olecranus* did not follow the standardized procedure used in subsequent studies. The latter procedure yields more reliable results, and its standardization provides that its results are directly comparable to the results of similar studies in other species. I therefore reinvestigated the range of motion at the shoulder in *M. olecranus*, using the latter procedure.

**Methods:** Casts of the left scapula and coracoid of *M. olecranus* were posed on a horizontal surface, supported from beneath with modeling clay, with the medial surface of the scapula facing toward the horizontal surface. A cast of the left humerus was posed at the limits of motion through the transverse and parasagittal planes. Photos of the poses in orthal views were superimposed and used to measure range of motion, which was measured as the angle between lines drawn down the long axis of the humerus in each position.

**Results:** Through the transverse plane, the humerus of *M. olecranus* could be elevated to a subhorizontal position and depressed to a subvertical position. It could move through the parasagittal plane from a subvertical position at full protraction to a position above the horizontal at full retraction. These results correct the previous mischaracterization of shoulder motion in *M. olecranus* as restricted to a small arc with the arms held in a permanent sprawl. The range of humeral motion in *M. olecranus* is much greater than that found by the previous method and allowed the animal to tuck its arms in at the sides, in addition to allowing them to sprawl so as to orient the palm downward. The wide range of humeral motion allowed *M. olecranus* to forage for insects by employing hook-and-pull digging at surfaces with a wider range of orientations than the previous study showed to be possible.

## INTRODUCTION

The primary literature of paleontology includes a plethora of studies in which previous studies are redone with improvements. For example, phylogenetic analyses are routinely

Corresponding author
Philip J. Senter, psenter@uncfsu.edu

redone with new data, often from newly discovered specimens, which enable each new phylogenetic analysis to be more informative than its previous counterparts. Likewise, morphological descriptions of fossil specimens are often redone when reexamination provides new insights (*e.g. Carpenter, Miles & Cloward, 2005*; *Carrano, Hutchinson & Sampson, 2005*; *Kobayashi & Barsbold, 2005*; *Turner, Makovicky & Norrell, 2012*). Similarly, investigations of the functional morphology of fossil taxa are often subject to restudy with improvement. A salient example of this is a series of investigations of forelimb posture in ceratopsid dinosaurs, in which subsequent studies introduced improvements that increased understanding and corrected previous errors (*Johnson & Ostrom, 1995*; *Thompson & Holmes, 2007*; *Fujiwara, 2009*; *Rega, Holmes & Tirabasso, 2010*). Here, I apply the principle of restudy with improvement to the investigation of forelimb function in the theropod dinosaur *Mononykus olecranus*.

Several previous investigations of the paleobiology of theropod dinosaurs have addressed the range of motion (ROM) in the forelimb or some part of it. For such studies, it is advantageous to have a repeatable protocol for numerically measuring ROM at given joints, which provides rigor and makes the results of one ROM study directly comparable to that of another. Unfortunately, early studies of theropod forelimb ROM did not involve the use of such a protocol. In some early investigations of theropod forelimb ROM, researchers illustrated poses that theropod forelimbs could achieve but did not report numerical measurements of ROM at specific joints (*Osmólska & Roniewicz, 1969*; *Galton, 1971*; *Nicholls & Russell, 1985*; *Carpenter, 2002*; *Kobayashi & Barsbold, 2005*) or reported such measurements in only a few joints (*Gishlick, 2001*). In other early investigations of theropod forelimb ROM, researchers reported measurements of ROM at specific joints but did not describe the protocols used to produce the measurements (*Welles, 1984*; *Sereno, 1993*; *Senter, 2005*).

In a 2005 study of forelimb ROM in *Acrocanthosaurus atokensis*, a colleague and I introduced and described a repeatable set of protocols for measuring ROM at specific joints (*Senter & Robins, 2005*). This protocol, hereafter abbreviated ARP (for "*Acrocanthosaurus* ROM protocol"), was repeated with minor variations in several subsequent studies on forelimb ROM in theropods and other non-avian dinosaurs (*Senter & Parrish, 2005*, *2006*; *Senter, 2006a*, *2006b*, *2006c*, *2007*; *Bonnan & Senter, 2007*; *Senter & Sullivan, 2019*). The ARP can be summarized as follows. To investigate shoulder ROM, the scapula and coracoid are posed on a table top, supported from beneath such that the medial surface of the scapular blade lies in a plane parallel with the floor, so that the table top approximates the animal's sagittal plane. This is because previous work has determined that the costal surface of the scapular blade faced medially, not medioventrally, in non-avian dinosaurs and basal birds (*Senter, 2006b*; *Senter & Robins, 2015*). The humerus is positioned in articulation with the glenoid cavity while supported from beneath by horizontal rods clamped to vertical structures, and is photographed in orthal views while posed at the extremes of motion through the transverse plane and at the extremes of motion through the parasagittal plane. The photos are digitally superimposed, a line is drawn down the long axis of the humerus in each pose in each view, and ROM at the shoulder is measured as the angle between the lines. To investigate elbow ROM, the

humerus, radius, and ulna are posed in articulation on a table top and photographed with the elbow in full flexion and full extension. The photos are digitally superimposed, a line is drawn down the long axis of the humerus and the radius, and ROM at the elbow is measured as the angle between the lines. To investigate ROM within each finger, the metacarpal and phalanges of each finger are posed in articulation on a table top and photographed with each finger in full flexion and full hyperextension. The photos are digitally superimposed, a line is drawn down the long axis of each bone, and ROM at each finger joint is measured as the angle between the lines. At each forelimb joint, the edges of the articular surfaces are presumed to delineate the limits of motion. Thus far, the ARP has been applied mainly to real bones or casts in real, three-dimensional space. However, it is also applicable to digitized bones on a computer screen, as shown by similarly done measurements of finger and toe ROM in digitized bones of the megaraptorid theropod *Australovenator* (*White et al., 2015*, *2016*).

When the ARP is applied to the shoulder joint, the humerus is moved in a hinge-like manner through the parasagittal plane to measure its range of protraction (flexion) and retraction (extension) within that plane, and the humerus is moved in a hinge-like motion through the transverse plane to measure its range of elevation and depression within that plane. Previous studies have determined that the proximal limb bones of tetrapods may use all six degrees of freedom, the use of which may include such movements as translation and long-axis rotation (*Arnold, Fischer & Nyakatura, 2014*; *Kambic, Roberts & Gatesy, 2014*; *Manafzadeh & Padian, 2018*; *Demuth, Rayfield & Hutchinson, 2020*; *Manafzadeh & Gatesy, 2021*; *Richards et al., 2021*; *Wiseman et al., 2022*). Accordingly, when the ARP is applied to the shoulder joint to measure ROM through the parasagittal plane, the humerus has been rotated about its long axis into the position that yields the maximal ROM through the parasagittal plane. Likewise, when the ARP is applied to the shoulder joint to measure ROM through the transverse plane, the humerus has been rotated about its long axis into the position that yields the maximal ROM through the transverse plane.

Two previous studies applied the ARP to species for which forelimb ROM had previously been illustrated or measured without the ARP. Forelimb ROM in *Deinonychus antirrhopus* had previously been studied by *Gishlick (2001)* and *Carpenter (2002)* and was later restudied with the ARP (*Senter, 2006a*). Forelimb ROM in *Dilophosaurus wetherilli* had previously been studied by *Welles (1984)* and was later restudied with the ARP (*Senter & Sullivan, 2019*). In both cases, restudy with the ARP provided new insights into the paleobiology of the animals (*Senter, 2006a*; *Senter & Sullivan, 2019*). Here, I report a restudy of shoulder ROM in the theropod *Mononykus olecranus*.

*Mononykus olecranus* is a small theropod dinosaur from Mongolia (*Perle et al., 1994*; *Chiappe, Norrell & Clark, 1996*). It is a member of the Late Cretaceous family Alvarezsauridae, which is part of the superfamily Alvarezsauroidea, early members of which are known from the Middle and Late Jurassic (*Choiniere et al., 2010*, *2013*; *Qin et al., 2019*). *M. olecranus* and other Late Cretaceous alvarezsauroids are notable for their highly derived forelimbs, which are extremely short, exhibit an elongate olecranon process on the ulna, and end with a hand in which the second and third fingers, when present, are strongly reduced in comparison to the thumb, which bears a prominent ungual phalanx

(*Perle et al., 1994*; *Novas, 1997*; *Suzuki et al., 2002*; *Longrich & Currie, 2009*; *Xu et al., 2011*; *Averianov & Sues, 2017*, *2022*; *Fowler et al., 2020*; *Freimuth & Wilson, 2021*; *Averianov & Sues, 2022*). In a previous study, forelimb ROM was measured in *M. olecranus* and found to be consistent with hook-and-pull digging, such as extant anteaters and pangolins use to break into tough insect nests (*Senter, 2005*). That study was done before the ARP was introduced. The measurements of ROM in the elbow and finger of *M. olecranus* were accomplished as in the ARP, except that a flatbed scanner was used in lieu of both the table top and the camera. For these joints, this method was sufficiently similar to the ARP for its results to be directly comparable to those achieved with the ARP, because in both cases, the ROM of hinge joints was measured in the plane of motion.

Unlike theropod elbow and finger joints, the shoulder joint in theropods is not a hinge joint but instead allows motion in three dimensions. This creates a challenge for posing bones that is solved by the ARP but was problematic for the 2005 study of shoulder ROM in *M. olecranus*. In that study, casts of the scapula, coracoid, and humerus of *M. olecranus* were suspended from a chemistry ring stand with stiff, insulated electrical wire, with the medial surface of the scapula in a vertical plane instead of a horizontal plane (*Senter, 2005*). Although that method allowed the illustration of humeral movement in three dimensions, the stiff wires introduced restriction of movement, as well as visual obstruction of the small specimen, which later led me to question whether I had correctly judged the limits of motion in *M. olecranus*. Subsequent experience with the ARP reinforced that suspicion. It therefore seemed prudent to repeat the study of the shoulder ROM of *M. olecranus*, using the ARP.

The study reported here involved manual manipulation of real objects (cast of bones) in real space. In recent years, digital techniques have become available that enable the digital use of images on a computer screen to perform equivalent or related tasks, and some such techniques have provided useful insights into limb motion in animals (*Thompson & Holmes, 2007*; *Mallison, 2010*; *Pierce, Clack & Hutchinson, 2012*; *Baier & Gatesy, 2013*; *Arnold, Fischer & Nyakatura, 2014*; *Kambic, Roberts & Gatesy, 2014*, *2017*; *White et al., 2015*, *2016*; *Manafzadeh & Padian, 2018*; *Demuth, Rayfield & Hutchinson, 2020*; *Demuth et al., 2023*; *Manafzadeh & Gatesy, 2021*; *Richards et al., 2021*; *Wiseman et al., 2022*; *Dempsey et al., 2023*). A computerized equivalent of the ARP can be accomplished with digital manipulation of three-dimensional images of scanned bones on a computer screen. When applied to theropod forelimb bones (*White et al., 2015*), this produces results that are similar to those found in other theropod species with similar forelimb features by manipulation of real casts or bones in real space (*e.g. Senter, 2006a*, *2006c*; *Senter & Parrish, 2006*). Similarly, when the ARP is adapted to the theropod foot, digital manipulation of images on a computer screen (*White et al., 2016*) produces results similar to those produced by physical manipulations of bones or casts (*Senter, 2009*). Digital and physical manipulation therefore seem to produce results of equivalent quality, so that there is currently no justification to favor one method over the other, except in special cases. Such cases occur when digital techniques are necessary to overcome logistical difficulties in the use of real bones or casts. For example, if fossil limb bones remain embedded in matrix, three-dimensional scans of them can be used to measure ROM by digital manipulation

(*Pierce, Clack & Hutchinson, 2012*). In the study reported here, no such logistical difficulties were present, so digital manipulation was not necessary.

## MATERIALS AND METHODS

For this study, I manually manipulated casts (YPM 56693) of the left scapula, coracoid, and humerus of the holotype of *Mononykus olecranus* (GI N107/6) to determine humeral ROM, using the ARP. First, I positioned the scapula and coracoid upon the horizontal surface of a wooden box, with the medial surface of the scapula facing downward, so that the horizontal surface of the box approximated the animal's midsagittal plane. The two bones were held together in articulation with a narrow strip of masking tape and were supported from beneath by a blob of modeling clay, which allowed the anterior scapula and coracoid to curve toward the table, just as they would have curved medially in the living animal (Fig. 1A). I posed the humerus at the extremes of motion through the parasagittal plane and photographed the poses in lateral view. I then posed the humerus at the extremes of motion through the transverse plane and photographed the poses in cranial (anterior) view. Throughout the process, the long axis of the scapular blade was oriented at 21° to the edge of the wooden box, in accordance with the findings of *Senter & Robins (2015)*, based on articulated specimens, that the theropod scapula was oriented with the long axis of its blade at approximately 21° from the sacrum, which was approximately parallel with the animal's horizontal craniocaudal (anteroposterior) axis. This allowed a line parallel to the edge of the box to serve as a proxy for the animal's horizontal craniocaudal axis during photography.

As in previous studies using the ARP, the photographic images were digitally superimposed, a line was drawn down the long axis of the humerus in each position, and ROM was measured as the angle between the lines. In cranial view, the shaft of the humerus is straight, and the line representing its long axis was drawn down the middle of the humerus, parallel with the shaft. In lateral view, the humerus is curved (convex posteriorly); the line representing its long axis was drawn as a line connecting the center of the humeral head with the center of the lateral epicondyle.

There was no need to support the cast of the humerus from beneath with horizontal rods, because the use of modeling clay provided a unique opportunity to support the cast in a different and simpler way. The humerus was held in place in all but one position by gently pressing the internal tuberosity into the clay. The humerus of *M. olecranus* has a strongly projecting internal tuberosity, which made it easy to pose in this manner. The one position in which the internal tubercle was not pressed into the clay was full elevation of the humerus through the transverse plane, in which position the internal tubercle was clear of the clay. In this position, the humerus was held in place by gently pressing the humeral head onto the clay to create a shallow, form-fitting depression that supported the humerus from beneath.

The humerus of *M. olecranus* does not fit precisely into the glenoid cavity (Fig. 1B). Such is also the case in extant archosaurs, in which there is a cap of articular cartilage with a shape that does not precisely match that of the underlying bony surface of the glenoid (*Holliday et al., 2010*). In accordance with the extant phylogenetic bracket approach

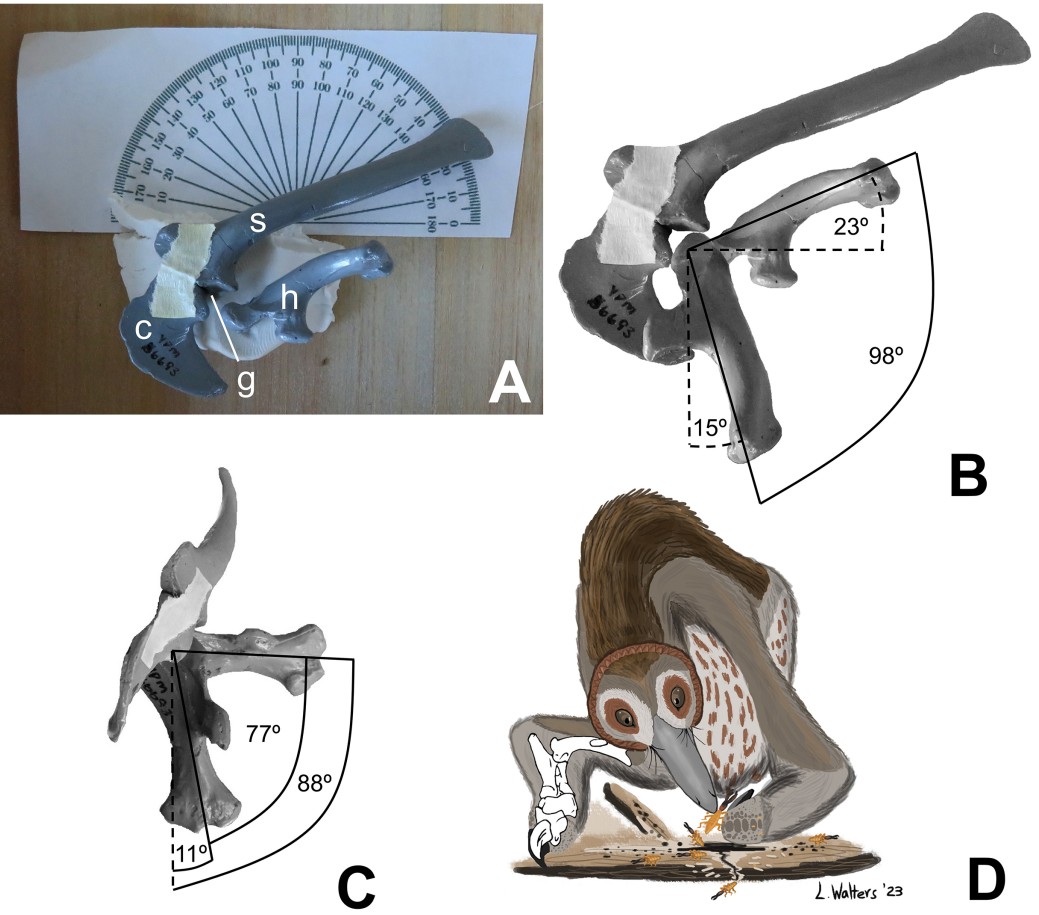

**Figure 1 Range of motion in the shoulder of *Mononykus olecranus*, and life reconstruction.** (A) Experimental setup, photographed from above. (B) Range of humeral motion through the parasagittal plane, in lateral view. (C) Range of humeral motion through the transverse plane, in anterior (cranial) view. (D) Life reconstruction of *Mononykus olecranus* foraging for termites in some wood. Note that the right forelimb is in a sprawl, whereas the left forelimb has the elbow tucked in at the side, to show that the forelimb of *Mononykus* was capable of both poses. The facial disk in this reconstruction is based on that of the barn owl (*Tyto alba*), in light of the presence of sensory anatomy in alvarezsaurids that suggests nocturnal foraging and a sense of hearing comparable to that of the barn owl (*Choiniere et al., 2021*).

(*Witmer, 1995*), it is reasonable to infer that a homologous cartilage cap was present in *Mononykus*, accounting for the small gap between the opposing bony articular surfaces of the humeral head and the glenoid cavity through the entire ROM. However, the glenoid cavity and the head of the humerus are sufficiently well defined to allow the ARP to be used as it was in previous studies of theropod shoulder ROM (*Senter & Robins, 2005*; *Senter, 2006a*, *2006b*, *2006c*, *2007*; *Senter & Parrish, 2006*; *Bonnan & Senter, 2007*; *Senter & Sullivan, 2019*).

To estimate the influence of the articular cartilage cap and other soft tissues upon ROM, it is useful to compare ROM as found with bare bones to ROM as found with an intact animal. Shoulder ROM in intact cadavers of the American alligator (*Alligator mississippiensis*), in all dimensions, is similar to but slightly greater than that found with

bare bones (*Hutson & Hutson, 2013*). Because the alligator represents one the two major lineages of extant archosaurs, one can tentatively infer that the same was the case for *Mononykus olecranus*, another archosaur. This inference cannot be drawn with complete certainty, because the relationship between *Mononykus* and extant archosaurs is distant. However, if it is correct, the shoulder ROM as found here with bare bones can be considered a close approximation of the ROM in the shoulder of the live animal, which is expected to be only slightly greater.

## RESULTS AND CONCLUSIONS

Application of the ARP to shoulder ROM in *M. olecranus* confirmed that the method used to estimate shoulder ROM in the previous study (*Senter, 2005*) was inadequate. Humeral ROM as found here is much greater than that reported in the previous study. In the present study, I found that the humerus can be swung in the parasagittal plane through an arc of 98°, from a subvertical position (15° from the vertical, in lateral view) at full protraction (shoulder flexion) to a position in which the elbow is higher (more dorsal) than the glenoid cavity (with the humerus at 23° above the horizontal, in lateral view) at full retraction (shoulder extension) (Fig. 1B). The humerus of *M. olecranus* can be swung in the transverse plane through an arc of 77°, from a subvertical position (11° from the vertical, in cranial view) to a subhorizontal position (88° from the vertical, in cranial view) (Fig. 1C). If the situation in alligators—in which shoulder ROM in intact cadavers is only slightly greater than that found with bare bones (*Hutson & Hutson, 2013*)—applies to *Mononykus*, then the range of motion in the humerus of live *M. olecranus* was similar to but slightly greater than the measurements listed above.

## DISCUSSION

Humeral ROM in *M. olecranus* is much greater than previously reported (*Senter, 2005*). The previous study correctly found that humeral motion through the transverse plane allowed the humerus of *M. olecranus* to be raised to a subhorizontal position, parallel with the ground (*Senter, 2005*). However, unlike in the previous study, here I find that the humerus can additionally be depressed through the transverse plane to a subvertical position. In the previous study, misjudgment of shoulder ROM led to the conclusion that humeral motion through the parasagittal plane was limited to an arc of only 23°, with neither a subvertical position achievable at maximum protraction nor a subhorizontal position achievable at maximum retraction (*Senter, 2005*). In contrast, here I find that humeral motion through the parasagittal plane is more extensive, allowing protraction to a subvertical position, and allowing retraction to a subhorizontal position and beyond.

Inability to elevate the humerus to a subhorizontal position through the transverse plane appears to be plesiomorphic for dinosaurs. ROM studies show that this inability is shared among basal theropods (*Carpenter, 2002*; *Senter & Robins, 2005*; *Senter & Sullivan, 2019*), sauropodomorphs (*Bonnan & Senter, 2007*), and basal ceratopsians (*Senter, 2007*). Within the Dinosauria, the ability to elevate the humerus to a subhorizontal position through the transverse plane was acquired independently at least four times: in derived Ceratopsia (*Senter, 2007*), in the theropod clade Ornithomimosauria (*Senter, 2006b*), in the

theropod clade Paraves (*Senter, 2006a*), and in *Mononykus* and its kin. In basal ceratopsians and in the clades Ornithomimosauria and Paraves, this ability was acquired *via* an extension of the glenoid cavity onto the lateral surface of the scapulocoracoid, enabling the humeral head to remain in articulation with the glenoid cavity through a greater arc of motion (*Senter, 2006a*, *2006b*, *2007*). In *M. olecranus*, there is no such glenoid extension, and the ability was achieved differently. It was accomplished *via* an offset humeral head, which is not in line with the humeral shaft (as it is in other theropods) but is shifted toward the deltopectoral crest (*Senter, 2005*). Such is also the case in the humerus of other alvarezsaurids (*Chiappe, Norrell & Clark, 1998*; *Averianov & Sues, 2022*) and in the Late Cretaceous alvarezsauroid *Patagonykus* (*Novas, 1997*). In contrast, the humeral head of basal alvarezsauroids from the Late Jurassic and Early Cretaceous is in line with the humeral shaft, as in non-alvarezsauroid theropods (*Choiniere et al., 2010*; *Xu et al., 2018*; *Qin et al., 2019*). It therefore appears that the ability to elevate the humerus through the transverse plane into a subhorizontal position is a property of the same alvarezsauroid clade in which the hand is modified such that there is one large finger with a prominent claw, and the other fingers are reduced or absent.

Pangolins, anteaters, and some armadillos have hands in which there is one large finger with a prominent claw, and the other fingers are much smaller. Here, I coin the term skalodactyly (from the Greek *skalis daktylos*: shovel finger) for this condition and the term skalodactyl for the specialized finger with the prominent claw. Extant skalodactylous animals use the skalodactyl for hook-and-pull digging, a means to crack into plants and tough insect nests to gain access to ants and termites within (*Walker et al., 1975*; *Montgomery, 1983*; *Redford, 1985*; *Taylor, 1985*; *Rahm, 1990*). Skalodactyly is therefore related to myrmecophagy (a diet of ants and/or termites) in extant animals. Late Cretaceous alvarezsauroids are skalodactylous, which suggests similar forelimb use, hence a similar diet (Fig. 1D). The ability of Late Cretaceous alvarezsauroids to elevate the humerus, thereby orienting the palm downward (ventrally), is plausibly related to such behavior, as it would have facilitated cracking into horizontal surfaces, such as the top of an insect nest or a fallen log.

The interpretation of alvarezsaurids as insectivores is consistent with their dental morphology. In most other theropods, the teeth are larger, serrated, and recurved, a morphology that suggests a diet of vertebrate prey. In contrast, alvarezsaurid teeth are smaller, more numerous, unserrated, and not recurved (*Perle et al., 1994*; *Chiappe, Norrell & Clark, 1998*), indicating a dietary shift. The teeth of basal alvarezsauroids from the Jurassic Period are serrated and recurved (*Choiniere et al., 2010*, *2013*), suggesting that this dietary shift had not yet occurred in the basal members of Alvarezsauroidea.

Some authors have expressed doubt that alvarezsaurids were myrmecophagous (*Alifanov & Saveliev, 2011*; *Agnolín et al., 2012*; *Averianov, Skutchas & Lopatin, 2023*). However, alvarezsaurids exhibit a suite of traits consistent with myrmecophagy in addition to skalodactyly: an anterior dentary diastema; simplified teeth that are reduced in size; long, narrow jaws with weak mandibles; and reduced jaw articulations (*Longrich & Currie, 2009*). Each of these traits is also found in some non-myrmecophagous animals, *e.g.* manatees (anterior dentary diastema), vampire bats (reduced and simplified teeth, reduced

jaw articulation), and avocets (long, narrow jaw with weak mandible). However, the presence of the entire suite of traits together is strongly indicative of myrmecophagy. *Averianov, Skutchas & Lopatin (2023)* posited that the reduced forelimbs of alvarezsaurids were too small to have been used to break into termite mounds. However, the forelimb of *M. olecranus* is similar in absolute size to that of the silky anteater (*Cyclopes didactylus*), which uses its skalodactyl to break into twigs and woody vines to feed on the insects within (*Montgomery, 1983*). The similar size of the forelimb in *M. olecranus* would therefore not have prohibited such use. *Averianov, Skutchas & Lopatin (2023)* also argued that extant myrmecophagous animals lack the finger reduction that alvarezsaurids have. However, that is incorrect. The fingers other than the skalodactyl are reduced, and in some cases are even vestigial, in some extant myrmecophagous animals with skalodactyly: the giant armadillo (*Priodontes maximus*) and all extant species of anteaters (*Myrmecophaga tridactyla*, *Tamandua* spp., and *Cyclopes didactylus*) (*Senter & Moch, 2015*) (Fig. 2). *Alifanov & Saveliev (2011)* posited that the alvarezsaurid forelimb was functionally analogous to that of the ostrich, implying that its small size was due to loss of function. However, the alvarezsaurid forelimb exhibits a suite of traits that are more suggestive of a specialized novel function: an offset humeral head (which increased the range of transverse motion), an elongated olecranon process (which increased the leverage of the arm extensor musculature), a deeply ginglymoid articulation between the penultimate phalanx and ungual of the thumb, and a much greater range of motion in the ungual phalanx of the thumb (*Senter, 2005*) than is present in the ungual phalanges of other theropods (*Osmólska & Roniewicz, 1969*; *Galton, 1971*; *Nicholls & Russell, 1985*; *Gishlick, 2001*; *Carpenter, 2002*; *Kobayashi & Barsbold, 2005*; *Senter & Parrish, 2005*; *Senter & Robins, 2005*; *Senter, 2006a*, *2007*; *Senter & Sullivan, 2019*).

Not all myrmecophagous animals are skalodactylous. Some are scratch-diggers with fingers of more uniform size. Examples include aardvarks (*Orycteropus afer*), sloth bears (*Melursus ursinus*), and armadillos other than *Priodontes*. However, all extant skalodactylous animals are myrmecophagous hook-and-pull diggers, and skalodactyly has no known functional significance other than hook-and-pull digging as a part of myrmecophagous behavior. The inference that alvarezsaurids were myrmecophagous is therefore supported by their forelimb morphology.

The use of the specialized hands of Late Cretaceous alvarezsauroids for hook-and-pull digging does not preclude their use to crack into eggshells. Some previous researchers have hypothesized that alvarezsaurids fed on eggs, because one specimen was associated with a broken shell from an egg that was too big for it to have laid, and another was associated with two eggs (*Lü et al., 2018*). In the latter case, egg size is consistent with the alvarezsaurid as the parent, but a lack of nest structure makes the possible alvarezsaurid parentage uncertain (*Agnolín et al., 2012*). It is possible that the alvarezsaurid skalodactyl was used to break into eggs, but this hypothesis should be viewed with caution until further support is obtained, because there is no extant precedent for such use of a skalodactyl, and the association of one fossil specimen with a piece of fossil eggshell may have been accidental.

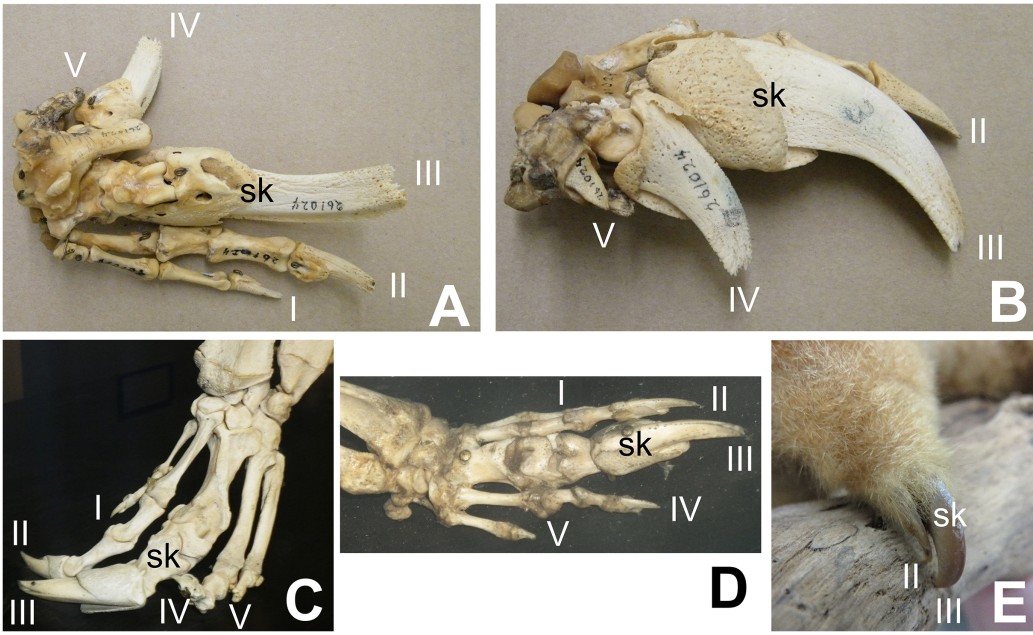

**Figure 2** **Digital proportions in skalodactylous mammals, showing reduction of fingers other than the skalodactyl.** (A and B) Right hand skeleton of giant armadillo (*Priodontes maximus*) in palmar view (A) and lateral view (B), showing vestigialization of digit V and strong reduction of digit IV. (C) Left hand skeleton of giant anteater (*Myrmecophaga tridactyla*) in dorsal view, showing vestigialization of finger V and strong reduction of digit I. (D) Right hand skeleton of southern tamandua (*Tamandua tetradactyla*) in dorsal view, showing vestigialization of digit V. (E) Left hand of silky anteater (*Cyclopes didacylus*), in which digits I and V are lost, and digit IV is vestigial.

The range of humeral motion through the parasagittal plane in *M. olecranus* resembles that of other theropods. In theropods generally, parasagittal movement of the humerus allows it to be protracted to a subvertical position and retracted to a subhorizontal position or further, with the elbows tucked in at the sides (*Senter & Robins, 2005*; *Senter, 2006a*; *Senter & Sullivan, 2019*). The ability to keep the elbows tucked in at the sides during protraction and retraction would have been advantageous, minimizing the tendency to snag the forelimbs on vegetation, which would have been a problem if the forelimb posture were restricted to a sprawl. Additionally, the ability to tuck the elbows in at the sides would have facilitated hook-and-pull digging into surfaces that were diagonal or vertical (for example, upright stems of woody plants), instead of restricting the digging behavior to horizontal surfaces. Thus, this provided the animal with a wider range of possible foraging stations than a permanent sprawl would have allowed.

This restudy of shoulder ROM in *Mononykus olecranus* brings the study of forelimb ROM in *M. olecranus* into alignment with the protocols used in previous studies that used the ARP, thereby ensuring that measurements of shoulder ROM in this species are directly comparable to those found for other species. This investigation also confirms that humeral motion in *M. olecranus* was greater than previously known. It therefore elucidates aspects of the paleobiology of *Mononykus olecranus* that were previously obscured by the mistaken characterization, from the prior study (*Senter, 2005*), that the alvarezsaurid humerus was

held in a permanent sprawl. Thus, an unfortunate and longstanding previous error in the study of theropod paleobiology is now corrected.

## INSTITUTIONAL ABBREVIATIONS

**GI**      Geological Institute, Mongolian Academy of Sciences, Ulaanbaatar, Mongolia
**YPM**    Yale Peabody Museum, New Haven, Connecticut, USA

## ANATOMICAL ABBREVIATIONS

**I–V**    first through fifth digits
**c**       coracoid
**g**       glenoid cavity
**h**       humerus
**s**       scapula
**sk**      skalodactyl

## ACKNOWLEDGEMENTS

For help with this study, part 19 of the Dinosaur Forelimb Project, I thank the following people. Daniel Brinkman (Yale Peabody Museum) provided the loan of the cast skeleton of *Mononykus*. Leandra Walters provided the life reconstruction of *Mononykus* in Fig. 1D. Nicole Edmison, Charles Potter, and Linda Gordon (United States National Museum) provided access to the specimen illustrated in Figs. 2A and 2B. Eileen Westwig (American Museum of Natural History) provided access to the specimens illustrated in Figs. 2C and 2D. Lisa Gatens and Benjamin Hess (North Carolina Museum of Natural Sciences) provided access to the specimen illustrated in Fig. 2E. Denver Fowler and an anonymous reviewer provided helpful feedback, which resulted in improvements to this article.

### Funding

The author received no funding for this work.

### Competing Interests

The author declares that they have no competing interests.

### Author Contributions

- Philip J. Senter conceived and designed the experiments, performed the experiments, analyzed the data, prepared figures and/or tables, authored or reviewed drafts of the article, and approved the final draft.

### Data Availability

There are no raw data and there is no code.

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
