# Peer review of "Restudy of shoulder motion in the theropod dinosaur Mononykus olecranus (Alvarezsauridae)"

_PeerJ, doi:10.7717/peerj.16605_

## Round 0.1 · original submission · Major Revisions

The reviewers have constructive critiques for the paper. One is correct that more literature review of digital methods for ROM/joint mobility assessment in the past 5 years is essential. The methods should be defended vs. these newer standards. While true that a manual method is more accessible/easy, that does not mean it is a reliable method scientifically, so this needs better backing up (objectively, are the methods and data good enough to support the conclusions?). Some added figures as per reviewer 1 are a good idea.

Other key papers:
https://journals.biologists.com/jeb/article/226/Suppl_1/jeb245042/286770/Joint-mobility-as-a-bridge-between-form-and
https://academic.oup.com/icb/article/62/5/1369/6548895

·

Basic reporting

In this short manuscript, Senter presents a restudy of the range of motion of the aberrant theropod dinosaur Mononykus. The author chose to repeat a previous 2005 study, but using a new methodology of his own devising which is more accurate. The author found that ROM was much greater than previously thought, and notes the consequent effects on insectivorous foraging behavior.

The paper is well written and I suggest acceptance with very minor edits, or as is.

CITATIONS
I suppose in terms of completeness of alvarezsaurid literature, Fowler et al., 2020 (of which I am first author) and Freimuth et al 2021 papers might be included since they describe forelimb material of the Hell Creek alvarezsaurid, Trierarchuncus, including the most complete manual ungual D-I (which demonstrates that curvature is greater than typically illustrated, with significant bearing on function). I suppose it isn’t necessary, but these papers contribute as much as the cited Averianov papers at least.

FIGURES
The single figure is fine. However, since the purpose of the paper is to use a repeatable methodology, then it might be nice to have a photo of the author’s physical setup. This is described in the methods, but without a photo. This isn’t necessary to add, but might help future researchers use the author’s method. I know such setups can look a bit like a home DIY project, with rubber bands and bits of wood everywhere (ours certainly did), but it is helpful to see something instead of just working from the description.

Experimental design

METHOD
I know some people have been working on computational ROM methods, and they might think that this old-school arrangement of casts is a little outdated. I am not in agreement with this perspective. Arrangement of casts (or real bones) has its issues (indeed, that is the purpose of this paper after all), but it leads to genuine insight and is readily available as a method, requiring little advanced technology.

Validity of the findings

DISCUSSION SUMMARY
I am in general agreement with the conclusions of this paper, so I don’t have any objections to alvarezsaurs as myrmecophages, and I welcome the discussion, especially the parts about anteaters etc. I suppose I do not know if we need a new word for their particular arrangement of the forelimb, but I can see the use for specifying organisms which have this single enlarged digit.

Additional comments

LINE SUGGESTIONS
LINE 253: Although each of these traits is found in some non-myrmecophagous animals (Agnolín et al., 2012),

Could you give some examples of these non-myrmecophagous animals please?

LINES ~250-265: Discussion of various anteaters etc.

Since comparisons are being made between alvarezsaurids and these extant myrmecophages, it might be nice to have some line drawings of their forelimbs (E.g. the line drawings in Senter 2005), especially since (lines 260-265) the author suggests that previous objections to these comparisons (based on phalanx length) are incorrect.

LINE 270: However, the alvarezsaurid forelimb exhibits a suite of traits that are more suggestive of a specialized novel function: an offset humeral head (which increased the range of transverse motion), an elongated olecranon process (which increased the leverage of the arm extensor musculature), and an ungual phalanx with a much greater range of motion (Senter 2005) than is present in the ungual phalanges of other theropods

You might consider adding in the clearly derived, deeply ginglymoid articulation of the D-I ungual. This is indicative of restricting range of motion to a single plane and extreme strengthening of the joint (seen to a much lesser extent in the feet of Dromaeosaurs, especially D-II; see Fowler et al 2011).


Denver Fowler

Reviewer 2 ·

Basic reporting

The manuscript by Senter re-investigates the shoulder mobility in the alvarezsaurid Mononykus using a standardized approach from a previous study (Senter & Robins 2005) and finds a larger range of motion (ROM) than previously thought allowing it to forage actively and more effectively with the forelimbs.

The figure is appropriate for the paper.

Some recent literature concerning methods should be discussed in the appropriate section. See below for references.

Due to comments and clarifications needed in the methods section and/or discussion thereof I recommend major revision.

Experimental design

While it is great to see more standardized approaches in the study of joint ROM, I have a few questions and comments on the applied methodology. There are several short-comings and limitations that need to be discussed in the manuscript. The mobility of Mononykus appears to be only measured around a single degree of freedom (DOF) at any time. The effect of simultaneously occurring motion in multiple DOFs is not addressed. Please have a look at Kambic et al. 2017 and Richards, et al., 2021. Furthermore, previous studies have found that in single DOF estimates the starting orientation of the bones have a large influence of the estimated joint mobility. The maxima and minima in all three rotational DOFs might not be reached simultaneously, and the initial orientation of a bone can therefore influence the result of the joint mobility measurement, see Manafzadeh & Gatesy 2021 and especially their figure S1. Were the bones moved in multiple DOFs to maximize the motion in the assumed/measured planes (see Richards et al., 2021) or was the humerus moved in a hinge-like fashion for each measured DOF, e.g., were the condyles always parallel in lateral view or was long-axis rotation allowed? If not, why not?

The articulation of the glenoid also appears somewhat unclear and needs further clarification and justification. Why was it assumed that the scapula and coracoid build a flat medial surface, flush with the box? I understand that both bones were resting on the wooden box and where then taped to hold them together, is this correct? Was there no curvature of the scapulocoracoid medially to allow it to wrap around the ribcage? It appears that this is somewhat the case in figure 1C, but it is not sufficiently described. The mediolateral orientation and angle of the coracoid respective to the scapula and both of them relative to the humerus could have influenced the ROM estimate, especially for a single DOF measurements as mentioned above. Please elaborate on this.

Soft tissues, both cartilaginous as well as ligaments and muscles, can have a large influence on joint mobility, see Arnold, et al., 2014, and bare bones manipulation usually overestimates this mobility. Different cartilage thicknesses can have an influence on joint ROM and resulting inferences, see Demuth et al., 2020, Wiseman et al., 2021. Although this might be less problematic in the study presented here as the joint centre appears to be dynamic rather than fixed, i.e., it can slide. The cited alligator study is a best-case example where the difference between osteological and cadaveric ROM was minimal, but this is unfortunately not always the case. Please elaborate why it is or is not appropriate to test different joint spacings in the glenoid of Mononykus.

Digital tools have facilitated and further standardised ROM estimation techniques in the last few years. The current (digital) gold standard being Manafzadeh & Padian, 2018 and her subsequent papers. What advantages does the ARP have over these digital methods? I assume speed and efficiency, maybe there are others? Please also elaborate on the short comings of either method, digital vs. analogue.

Validity of the findings

The results and the subsequent conclusions seem very sensible to me and make sense. It is good to see that a previous error in the literature is corrected. However, there are some caveats concerning the methodology that need discussed, as mentioned above. They should, however, have no to minimal influence on the results.

Additional comments

Minor Comments (by line #):

400: Mallison, 2010 is in the reference list but is not cited in the main text. Please incorporate the citation and discuss it where you see fit.

References:

Arnold, P., Fischer, M.S. and Nyakatura, J.A. (2014), Soft tissue influence on ex vivo mobility in the hip of Iguana: comparison with in vivo movement and its bearing on joint motion of fossil sprawling tetrapods. J. Anat., 225: 31-41. https://doi.org/10.1111/joa.12187
Kambic, R.E., Roberts, T.J. and Gatesy, S.M. (2017), 3-D range of motion envelopes reveal interacting degrees of freedom in avian hind limb joints. J. Anat., 231: 906-920. https://doi.org/10.1111/joa.12680
Manafzadeh, A. R. and Padian, K. (2018), ROM mapping of ligamentous constraints on avian hip mobility: implications for extinct ornithodirans. Proc. R. Soc. B Biol. Sci. 285, 20180727. https://doi.org/10.1098/rspb.2018.0727
Demuth, O.E., Rayfield, E.J. and Hutchinson, J.R. (2020), 3D hindlimb joint mobility of the stem-archosaur Euparkeria capensis with implications for postural evolution within Archosauria. Sci Rep 10, 15357. https://doi.org/10.1038/s41598-020-70175-y
Richards, H.L., Bishop, P.J., Hocking, D.P., Adams, J.W. and Evans, A.R. (2021), Low elbow mobility indicates unique forelimb posture and function in a giant extinct marsupial. J Anat, 238: 1425-1441. https://doi.org/10.1111/joa.13389
Manafzadeh, A.R. & Gatesy, S.M. (2021), Paleobiological reconstructions of articular function require all six degrees of freedom. Journal of Anatomy, 239, 1516–1524. https://doi.org/10.1111/joa.13513
Wiseman, A.L.A, Demuth, O.E., Pomeroy, E. and De Groote, I. (2022), Reconstructing Articular Cartilage in the Australopithecus afarensis Hip Joint and the Need for Modeling Six Degrees of Freedom, Integrative Organismal Biology, 4: obac031. https://doi.org/10.1093/iob/obac031

---

## Round 0.2 · Minor Revisions

The reviewer is satisfied, and the other one's concerns were dealt with adequately, so I am OK to proceed with this if the recommended reference by Tsai et al. is incorporated as a citation and amendment to the text as the reviewer notes; for better fealty to the literature (https://onlinelibrary.wiley.com/doi/full/10.1111/joa.13101 ). Congratulations.

Reviewer 2 ·

Basic reporting

No comment

Experimental design

I thank the author for including the additional figure 1A. It is helpful to better illustrate the experimental setup.

The author has made changes to the introduction, discussing the wider field of joint mobility and its recent developments. While I disagree with the statement in lines 102-103 that there is minimal sliding present in the hip or shoulder joints of reptiles (e.g, see Tsai et al. 2020 JANAT), I agree that its relevance to this study is minimal. Sliding of the joint is inherently taken into account during phyisical manipulation because the joint centre is dynamic and the bones are constantly kept in articulation. Translations are mostly an issue in digital manipulation, where the joint centre is fixed in space and they need to be incorporated to ensure joint contact. I am, therefore, statisfied with the justifications presented in the introduction and the clarification of the rotational planes for the experiments.

Validity of the findings

As in my previous review, the findings are sensible and I am glad that the author was able to correct a mistake in the literature.

Additional comments

I am satisfied with the revision of the manuscript as the author has addressed all my concerns and I am pleased to now recommend its acceptance.

---

## Round 0.3 · Minor Revisions

To ensure the integrity of the literature, as there are quite a bit of empirical data in vivo and ex vivo for joint mobility, I consulted with the prior reviewer and they responded as follows:

There is both translation and rotation (circumduction). In nature there is no such thing as a perfect ball-and-socket joint (the only joint type at which no translational sliding occurs), not even in the very spherical hominin hip [this is very well known from many experimental studies of human hip biomechanics]. While Tsai et al. 2020 do not specifically mention translational movement in their manuscript, I understand it as being implied. The femoral head of Alligator is subellipsoidal in shape and as such cannot produce a perfectly spherical joint motion (i.e., the different ellipsoidal radii result in different rotational centres depending on the rotation axes), which need to be offset by translational movement to retain joint contact (e.g., rolling of the joint). The constant contact between the femoral neck and the antitrochanter menisci can, therefore, only be explained via some translational sliding of the femoral head within the joint capsule. The entire convex surface of the proximal femur is in contact with the acetabulum during motion, therefore, no central pivot akin to a ball-and-socket joint is present in Alligator, and thus requires sliding during movement to maintain contact. I hope this clarifies my position and how I understand that paper. Additionally, Manafzadeh & Gatesy 2021 (J Anat) report translational sliding of around 0.8 cm in all axes in the alligator hip joint (their Table 2), supporting my argument.

Regardless, this study already accounts for translational degrees of freedom in the shoulder joint, in that the humerus and glenoid of the alvarezsaurid are in constant contact, thus inherently implying that they are accounted for.

Therefore, beside the removal of the misleading statement that no (or only minimal) sliding is present in reptilian hip or shoulder joints, no further changes to the manuscript are required.

Please make that basic amendment.

---

## Round 0.4 · accepted · Accept

Thank you for addressing the final reviewer's comment. I've assessed the revision and approve it. Congratulations on your acceptance for publication!